# The synergistic effects of diet and physical activity on sleep quality among Chinese adolescents: A comprehensive cross-sectional study

## Research Article

Chinese adolescents; dietary behaviors; interaction effects; poor sleep quality; physical activity

**Corresponding author:**
Jing Guo;
Email: jing624218@163.com

Binyang Huang, Ning Huang, Chen Chen and Jing Guo 

School of Public Health, Peking University, Beijing, China

## Abstract

This study investigated the independent and interactive effects of dietary behaviors and physical activity on poor sleep quality among 15,059 Chinese adolescents. Using a cross-sectional design, we assessed sleep quality (Pittsburgh Sleep Quality Index, PSQI), dietary habits, and physical activity. Logistic regression and interaction analysis were performed to examine associations, adjusting for covariates. The prevalence of poor sleep quality (PSQI score ≥ 7) was 9.72%. Seven healthy dietary behaviors were identified as protective (e.g., regular diet, abstaining from alcohol; ORs=0.49–0.56). While physical activity showed no independent association, limiting screen-based sedentary screen time(≤2h/day) reduced poor sleep odds by 31% (OR = 0.69). Two significant interactions emerged: abstaining from alcohol combined with limiting sugary beverages synergistically reduced the odds of poor sleep quality by 42% (OR = 0.58), whereas the combination of healthy dining out and high physical activity was associated with a 181% increased odds of poor sleep quality (OR = 2.81). While healthy dietary patterns are strongly associated with better sleep quality, the interplay between behaviors is complex, demonstrating both synergistic protective association and antagonistic outcomes. Findings highlight the need for integrated lifestyle interventions that account for behavioral interactions in promoting adolescent sleep quality.

## Impact statements

Adolescent sleep deprivation is a global public health crisis. Our study addresses this gap by analyzing how these behaviors combine to affect sleep in over 15,000 adolescents. We move beyond a simple checklist approach to reveal the complex trade-offs teens make in their packed daily schedules. The key finding is a paradoxical one: combining two "healthy" behaviors" – high-level physical activity and healthy dining out – was associated with a significantly higher risk of poor sleep. This suggests that for many teens, the day becomes a stressful "time crunch." Intense, late-day exercise, followed by a socially stimulating meal, can leave an adolescent too physiologically and mentally aroused to sleep well, regardless of the health benefits of each individual activity. This finding has broad, practical utility for multiple stakeholders. Public health bodies and governmental organizations can use this evidence to design more effective health campaigns that promote a balanced, 24-h lifestyle rather than just isolated good habits. For schools, this research can inform scheduling policies – for instance, advising against late-evening, high-intensity extracurriculars on school nights to allow for adequate wind-down time. Clinicians and parents can use this insight to provide more nuanced guidance. Finally, while our study was conducted in China, the underlying issue of time pressure and competing lifestyle demands is universal for adolescents in demanding academic environments worldwide. Therefore, our methodology of systematically analyzing behavioral interactions serves as a valuable model for researchers and policymakers seeking to understand and address other complex health challenges in young people globally.




## Introduction

Sleep problems are a critical public health concern affecting adolescent development worldwide, with substantial implications for physical function, cognitive function and emotional regulation (St-Onge et al., 2016). According to the American Academy of Sleep Medicine expert panel (Kansagra, 2020), adolescents require 8–10 h of sleep per night to support optimal health and development. Nevertheless, epidemiological studies demonstrate a global prevalence of sleep problems among teenagers (Dohnt et al., 2012), a problem that is particularly acute in East Asian populations, where cultural emphasis on academic achievement often exacerbates sleep deprivation. In mainland China, recent meta-analytic findings (Liang et al., 2021) reveal that the

prevalence of sleep disorders in this demographic is significantly higher than that in the general population, reaching as high as 26%. To address the health risks faced by adolescents, it is necessary to design targeted strategies aimed at improving sleep quality.

Lifestyle factors are critical modifiable determinants of sleep health, operating through complex neurobiological mechanisms that primarily interact with the two-process model of sleep regulation: the homeostatic sleep drive (Process S) and the circadian system (Process C) (Borbély, 2022). Physiologically, physical activity acts as a potent regulator of these processes. Ideally, high-intensity activity increases adenosine accumulation and energy expenditure, thereby enhancing homeostatic sleep pressure (Process S) and facilitating deep sleep (Tan et al., 2020). Furthermore, acting as a non-photic "zeitgeber" (time cue), appropriate physical activity entrains the circadian clock through thermoregulation and neuroendocrine modulation (Healy et al., 2021). However, specific behavioral patterns may disrupt this balance. For instance, sedentary behaviors – characterized by prolonged screen time – disrupt sleep continuity through melatonin suppression and increased psychological arousal (Lissak, 2018; Sampasa-Kanyinga et al., 2020). Moreover, emerging evidence suggests that the relationship between physical activity and sleep may not be strictly linear but inverted U-shaped. Recent meta-analyses indicate that while moderate doses benefit sleep, excessive physical exertion – particularly without adequate recovery – may induce physiological hyperarousal that negates sleep benefits (Wang et al., 2025; Zhang et al., 2025a).

Similarly, dietary patterns influence sleep physiology through metabolic and chronobiological pathways. From a circadian perspective, regular meal timing and breakfast consumption serve as critical peripheral zeitgebers, synchronizing metabolic clocks (*e.g.*, the liver clock) with the central pacemaker to promote sleep consistency (Wehrens et al., 2017; Foster, 2020). Metabolically, diet quality directly impacts sleep architecture. High-sugar diets trigger rapid glucose fluctuations and inflammatory responses, which can activate arousal systems and fragment sleep (Kruger et al., 2014; Chaput et al., 2018). Furthermore, substance-specific behaviors like alcohol and caffeine consumption act as direct sleep disruptors by altering neurotransmitter stability and suppressing rapid eye movement (REM) sleep (Smith et al., 2019; Khan et al., 2021). The convergence of physical activity and diet on shared pathways – including inflammatory cytokines, glucose metabolism and circadian entrainment – suggests their effects on sleep are interactive. For instance, the anti-inflammatory effects of physical activity might buffer the sleep-disrupting impact of a pro-inflammatory diet (Krueger, 2008).

While existing epidemiological research establishes discrete associations, the current literature exhibits three key limitations. First, most studies adopt reductionist approaches, examining physical activity and diet in isolation despite their interdependent effects on sleep. Second, research in China, while growing, has often been constrained by smaller sample sizes, limiting generalizability. Third, and most importantly, existing interventions focus on single lifestyle domains, ignoring the complexity of adolescent health behaviors.

Addressing these gaps requires a comprehensive framework that considers not only the direct effects of lifestyle behaviors but also their interactive pathways. Therefore, the present study employs a census survey design in a county of northeastern China to: (1) examine the independent associations of specific dietary and physical activity behaviors with poor sleep quality; (2) investigate the synergistic or antagonistic effects of combined lifestyle factors; and (3) identify nuanced behavioral targets to inform the development of evidence-based, integrated interventions to improve adolescent sleep quality.

## Methods

### *Study design*

This cross-sectional survey was conducted in a representative county-level administrative unit in northeastern China between October 12 and November 5, 2023, in collaboration with the local education bureau. A total of 15,398 middle high-school students were recruited from 18 middle schools. School staff received training to standardize the data collection procedures. Under the guidance of their teachers, the students independently completed the electronic questionnaires *via* a web-based survey platform (https://www.wjx.cn/app/survey.aspx). To ensure data quality, we excluded participants who completed the questionnaire in <600 s or failed logical consistency checks. As a result, 15,059 subjects were included in the final analysis.

All participants, as well as their parents and teachers, gave consent after being informed about the aims of the survey. The study was approved by the Peking University Biomedical Institutional Review Board.

### *Measurement*

#### *Sleep quality*

The Pittsburgh Sleep Quality Index (PSQI) (Buysse et al., 1989) was used to evaluate the sleep quality of the participants over the past month. The PSQI is a self-report questionnaire consisting of 18 items that generate seven component scores: subjective sleep quality, sleep latency, sleep duration, habitual sleep efficiency, sleep disturbances, use of sleep medication and daytime dysfunction. Each component is scored from 0 to 3, with higher scores indicating poorer sleep quality. Consistent with established cutoffs used in adolescent populations, a global score above 7 was used to classify participants as having poor sleep quality. The Cronbach's alpha reliability coefficient for the PSQI in this sample was 0.68. This value aligns with previous large-scale epidemiological studies in adolescents, reflecting the characteristic floor effect of the medication use component in nonclinical populations (Passos et al., 2017; Setyowati and Chung, 2021).

#### *Diet behaviors*

Dietary behaviors were assessed using eight dichotomous variables derived from the Dietary Guidelines for Chinese School-aged Children (2022) (Zhang et al., 2022): (1) *Regular diet*: Daily consumption of three main meals without skipping; (2) *Healthy breakfast*: Inclusion of cereal, protein and dairy in morning meals; (3) *Healthy snack intake*: Selection of nutrient-dense snacks (*e.g.*, fruits and nuts) over high-sugar alternatives; (4) *Healthy dining out*: Selection of balanced meal options (avoiding salty, sweet or oily profiles) during takeout or dining out, assessed by dietary content rather than frequency; (5) *Daily milk intake*: Consumption of ≥300 mL pasteurized milk daily; (6) *Sufficient water intake*: Daily water consumption ≥1,500 mL excluding beverages; (7) *Limited consumption of sugary beverages*: ≤1 serving of sugar-sweetened drinks per week; and (8) *Abstaining from alcohol*: No consumption of alcoholic beverages.

These variables were measured *via* self-report questions (*e.g.,* "Have you been eating regularly every day for the past week?" for regular diet) and validated against national dietary guidelines to ensure construct validity.

### Physical activity and sedentary behavior

Physical activity and sedentary behavior were measured using items based on the Physical Activity Guidelines for Chinese Children and Adolescents (Zhang et al., 2017). These questions included "How often do you exercise and participate in physical activities per week?" and "What is the intensity of your physical exercise?." To ensure conceptual precision and align with standardized terminology (Falck et al., 2022), we operationalized our variables to explicitly differentiate between general physical activity and domain-specific screen behaviors.

The variable "Physical activity level" was categorized into three levels according to the following definitions: (1) *High-level physical activity*: Participants were considered to have a high level of physical activity if they engaged in high-intensity exercise three to five times per week, with each session lasting more than 60 min. (2) *Moderate-level physical activity*: This level was defined as either engaging in moderate-intensity exercise three to five times per week, with each session lasting more than 30 min, or engaging in high-intensity exercise three to five times per week, with each session lasting between 30 and 60 min. (3) *Low-level physical activity*: If participants did not meet the criteria for high- or moderate-level physical activity, they were classified as having a low level of physical activity.

It is important to note that while physical education (PE) is mandatory in Chinese schools, recent empirical evidence indicates that the effective duration of moderate-to-vigorous physical activity in standard PE classes often falls below our threshold for "Moderate-level" activity (>30 min per session) (Zhou et al., 2025). Consequently, students relying solely on mandatory PE would typically fall into the "Low-level" category, ensuring that our higher categories effectively capture voluntary, discretionary exercise (Wu et al., 2023).

The variable "Screen-based sedentary behavior" was categorized as ≤2 h/day *versus* >2 h/day. It was defined as the total time spent on activities such as using mobile phones, computers or watching TV, not exceeding 2 h per day. This measurement was also incorporated into the survey through relevant questions to comprehensively evaluate the participants' lifestyle and its potential impact on sleep health.

These operational definitions of physical activity level and sedentary activities involving electronic devices were established to ensure the consistency and comparability of data analysis in the subsequent sections of this study.

### Covariates

Covariates include sociodemographic characteristics, such as gender (male/female), ethnicity (Han/minority), grade (7th/8th/9th grade), self-reported health conditions (healthy/unhealthy) and chronic disease (yes/no). Family conditions include family residence (urban/rural), parental education level (primary school or below/junior high school/senior high school/junior college or above), self-reported family economy (normal/wealthy), only child status (yes/no) and primary caregiver (father/mother or other).

### Statistical analysis

Descriptive statistics are conducted to describe the characteristics of participants with different sleep problems. Categorical variables were represented as frequencies and percentages. The chi-square test is used to assess differences in various sociodemographic characteristics among different sleep quality groups. To examine the association between general healthy dietary behaviors and the sleep problem, a series of logistic regression are conducted using sleep problem as the main outcome variable and each dietary behavior, physical activity as the predictor, while adjusting for covariates such as sex, school grade, residential area and other relevant factors. Odds ratios (ORs) and 95% confidence intervals (CIs) were derived.

To explore the interactive effects among lifestyle behaviors, we systematically tested pairwise interactions among the eight dietary behaviors, physical activity level and screen-based sedentary behavior (a total of 45 interaction models). All models were adjusted for the full set of covariates. All analyses were performed using Stata 17.0, with statistical significance defined as a two-sided *p*-value <0.05.

## Results

### Descriptive characteristics of the sample

Table 1 shows descriptive analysis results, and 15,059 participants were included. The sample was 54.2% male, and the majority (93.3%) were of Han ethnicity. The majority of the participants (84.49%) are not the only child in their family. Nearly all participants (96.47%) self-reported their health status as "healthy," while 10.38% reported having chronic diseases. Regarding family situation, 36.93% of the participants live in urban areas, and the remaining 63.07% live in rural areas; 68.09% of participants are primarily cared for by their mother, 21.66% by their fathers and 10.24% by other family members. And the majority of the parents of participants had attained an educational level of middle school (50.33%) or high school (29.44%). The majority of participants (87.67%) described their family economy as "normal," with the remaining 12.33% indicating it as "rich."

As the data collected by applying the PSQI scales, it showed that the prevalence of poor sleep quality (PSQI ≥7) was 9.72%. Regarding the diet behaviors, 68.26% of the participants report eating regularly every day. Of these, a significant portion, amounting to 45.54%, also ensures that they got breakfast, including a variety of nutrients. Among the participants, 51.05% can limit their snack intake, while only 10.01% have consumed an excessive amount of sugary beverages; 42.41% of the participants consume milk daily, yet only 17.03% meet the daily water intake recommendation. Most of the participants (76.00%) can dine out healthily. Most of the participants did not smoke (95.02%) or drink alcohol (92.83%). As for the physical activity, 87.18% reported engaging in low levels of physical activity, compared to 8.63% and 4.18% who kept engaging in moderate and high levels, respectively; 76.03% of the participants can keep their sedentary activities with electronic devices within a 2-h limit.

### Associations of dietary behaviors and physical activity with poor sleep quality

As shown in Table 2 and Figure 1, the results show the association of diet and physical activity with poor sleep quality. After controlling for covariates, seven out of the eight healthy dietary behaviors were identified as protective factors against poor sleep quality. It indicated that participants with these behaviors had a lower risk of poor sleep quality compared to those without. The

**Table 1.** Descriptive statistics for all variables (*N* = 15,059)

| | Sleep problem (threshold: 7 points) | | | | | | $\chi^2$ |
| | No | | Yes | | Total | | |
| | N | (%) | N | (%) | N | (%) | Significance |
|---|---|---|---|---|---|---|---|
| Gender | | | | | | | |
| Male | 7,464 | 54.9 | 694 | 47.4 | 8,158 | 54.17 | 29.932 |
| Female | 6,131 | 45.1 | 770 | 52.6 | 6,901 | 45.83 | *** |
| Ethnicity | | | | | | | |
| Han | 12,684 | 93.3 | 1,365 | 93.21 | 14,049 | 93.29 | 0.0079 |
| Minority | 911 | 6.7 | 99 | 6.79 | 1,010 | 6.71 | |
| Grade | | | | | | | |
| 7th grade | 4,695 | 34.53 | 349 | 23.84 | 5,044 | 33.49 | 79.968 |
| 8th grade | 4,406 | 32.41 | 494 | 33.74 | 4,900 | 32.54 | *** |
| 9th grade | 4,494 | 33.06 | 621 | 42.42 | 5,115 | 33.97 | |
| Family residence | | | | | | | |
| Urban | 5,077 | 37.35 | 485 | 33.13 | 5,562 | 36.93 | 10.0865 |
| Rural | 8,518 | 62.65 | 979 | 66.87 | 9,497 | 63.07 | ** |
| Only child | | | | | | | |
| No | 11,486 | 84.49 | 1,238 | 84.56 | 12,724 | 84.49 | 0.006 |
| Yes | 2,109 | 15.51 | 226 | 15.44 | 2,335 | 15.51 | . |
| Primary caregiver | | | | | | | |
| Father | 2,950 | 21.7 | 311 | 21.24 | 3,261 | 21.66 | 35.265 |
| Mother | 9,315 | 68.53 | 938 | 64.07 | 10,253 | 68.1 | *** |
| Other | 1,327 | 9.76 | 215 | 14.69 | 1,542 | 10.24 | |
| Highest parental education level | | | | | | | |
| Primary school or below | 327 | 2.41 | 61 | 4.17 | 388 | 2.58 | 32.913 |
| Junior high school | 6,791 | 49.95 | 788 | 53.83 | 7,579 | 50.33 | *** |
| Senior high school | 4,027 | 29.62 | 407 | 27.8 | 4,434 | 29.44 | |
| Junior college or above | 2,450 | 18.02 | 208 | 14.21 | 2,658 | 17.65 | |
| Self-reported family economy | | | | | | | |
| Normal | 11,939 | 87.82 | 1,263 | 86.27 | 13,202 | 87.67 | 2.932 |
| Rich | 1,656 | 12.18 | 201 | 13.73 | 1,857 | 12.33 | . |
| Chronic disease | | | | | | | |
| No | 12,401 | 91.22 | 1,095 | 74.8 | 13,496 | 89.62 | 383.194 |
| Yes | 1,194 | 8.78 | 369 | 25.2 | 1,563 | 10.38 | *** |
| Self-reported health conditions | | | | | | | |
| Bad | 327 | 2.41 | 205 | 14 | 532 | 3.53 | 521.618 |
| Health | 13,268 | 97.59 | 1,259 | 86 | 14,527 | 96.47 | *** |
| Regular diet | | | | | | | |
| No | 4,010 | 29.5 | 769 | 52.53 | 4,779 | 31.74 | 323.608 |
| Yes | 9,585 | 70.5 | 695 | 47.47 | 10,280 | 68.26 | *** |
| Healthy breakfast | | | | | | | |
| No | 7,127 | 52.42 | 1,074 | 73.36 | 8,201 | 54.46 | 233.604 |
| Yes | 6,468 | 47.58 | 390 | 26.64 | 6,858 | 45.54 | *** |

*(Continued)*

**Table 1.** (*Continued*)

| | Sleep problem (threshold: 7 points) | | | | | | $\chi^2$ |
|---|---|---|---|---|---|---|---|
| | No | | Yes | | Total | | |
| | N | (%) | N | (%) | N | (%) | Significance |
| Healthy snack intake | | | | | | | |
| No | 6,426 | 47.27 | 945 | 64.55 | 7,371 | 48.95 | 157.963 |
| Yes | 7,169 | 52.73 | 519 | 35.45 | 7,688 | 51.05 | *** |
| Healthy dining out | | | | | | | |
| No | 3,021 | 22.22 | 593 | 40.51 | 3,614 | 24 | 242.247 |
| Yes | 10,574 | 77.78 | 871 | 59.49 | 11,445 | 76 | *** |
| Daily milk intake | | | | | | | |
| No | 7,635 | 56.16 | 1,037 | 70.83 | 8,672 | 57.59 | 116.503 |
| Yes | 5,960 | 43.84 | 427 | 29.17 | 6,387 | 42.41 | *** |
| Sufficient water intake | | | | | | | |
| No | 11,255 | 82.79 | 1,240 | 84.7 | 12,495 | 82.97 | 3.419 |
| Yes | 2,340 | 17.21 | 224 | 15.3 | 2,564 | 17.03 | . |
| Limited consumption of sugary beverages | | | | | | | |
| No | 1,191 | 8.76 | 316 | 21.58 | 1,507 | 10.01 | 241.354 |
| Yes | 12,404 | 91.24 | 1,148 | 78.42 | 13,552 | 89.99 | *** |
| Abstaining from alcohol | | | | | | | |
| No | 801 | 5.89 | 279 | 19.06 | 1,080 | 7.17 | 344.105 |
| Yes | 12,794 | 94.11 | 1,185 | 80.94 | 13,979 | 92.83 | *** |
| Smoking | | | | | | | |
| No | 13,037 | 95.9 | 1,272 | 86.89 | 14,309 | 95.02 | 226.738 |
| Yes | 558 | 4.1 | 192 | 13.11 | 750 | 4.98 | *** |
| Physical activity level | | | | | | | |
| Low level | 11,798 | 86.78 | 1,331 | 90.92 | 13,129 | 87.18 | 21.047 |
| Moderate level | 1,216 | 8.94 | 84 | 5.74 | 1,300 | 8.63 | *** |
| High level | 581 | 4.27 | 49 | 3.35 | 630 | 4.18 | |
| Screen-based sedentary behavior (≤2 h) | | | | | | | |
| No | 3,015 | 22.18 | 595 | 40.64 | 3,610 | 23.97 | 247.247 |
| Yes | 10,580 | 77.82 | 869 | 59.36 | 11,449 | 76.03 | *** |
| *N* | 13,595 | 90.28 | 1,464 | 9.72 | 15,059 | | |

*$P < 0.05$; **$P < 0.01$; ***$P < 0.001$.
Having sleep problem is defined as PSQI ≥7.
Model A: Unadjusted for covariates.
Model B: Adjusted for covariates including residential area, gender, grade, ethnicity, smoking status, chronic disease status, self-rated health, primary caregiver and highest parental education level.

strongest protective factors were maintaining a regular diet (OR = 0.49, 95% CI: 0.43–0.55), abstaining from alcohol (OR = 0.53, 95% CI: 0.44–0.65) and limiting sugary beverage consumption (OR = 0.56, 95% CI: 0.48–0.65). Sufficient water intake was the only dietary behavior not significantly associated with sleep quality.

In the unadjusted model, moderate physical activity was associated with lower odds of poor sleep quality (OR = 0.74, 95% CI: 0.58–0.94). However, after adjusting for covariates, the associations for both moderate (OR = 0.81, 95% CI: 0.63–1.03) and high physical activity (OR = 0.94, 95% CI: 0.68–1.30) were no longer statistically

significant. In contrast, screen-based sedentary behavior (≤2 h) remained a robust protective factor, associated with a 31% reduction in the odds of poor sleep quality (OR = 0.69, 95% CI: 0.61–0.79).

### Interaction effects of diet and physical activity on poor sleep quality

After adjusting for covariates, two statistically significant interaction effects emerged (Table 3 and Figure 2). First, a synergistic protective interaction was found between abstaining from alcohol and limiting sugary beverage consumption. Compared to

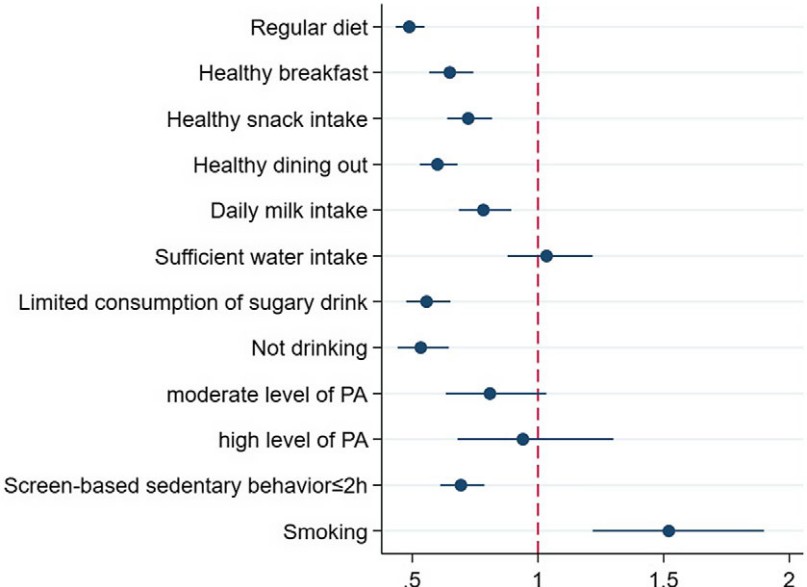

**Figure 1.** Distribution of odds ratios for poor sleep quality associated with dietary behaviors and physical activity (Model C).

**Table 2.** Association between dietary behaviors, physical activity and poor sleep quality (N = 15,059)

| | Model A OR (95% CI) | Model B OR (95% CI) |
|---|---|---|
| Regular diet | 0.46 [0.41,0.51]*** | 0.49 [0.43,0.55]*** |
| Healthy breakfast | 0.61 [0.54,0.70]*** | 0.65 [0.57,0.74]*** |
| Healthy snack intake | 0.69 [0.62,0.78]*** | 0.72 [0.64,0.82]*** |
| Healthy dining out | 0.58 [0.51,0.65]*** | 0.60 [0.53,0.68]*** |
| Daily milk intake | 0.72 [0.64,0.82]*** | 0.78 [0.69,0.89]*** |
| Sufficient water intake | 1.03 [0.88,1.21] | 1.03 [0.88,1.22] |
| Limited consumption of sugary beverages | 0.57 [0.49,0.66]*** | 0.56 [0.48,0.65]*** |
| Abstaining from alcohol | 0.40 [0.34,0.47]*** | 0.53 [0.44,0.65]*** |
| Physical activity level (Ref: low level) | | |
| Moderate level | 0.74 [0.58,0.94]* | 0.81 [0.63,1.03] |
| High level | 0.80 [0.59,1.09] | 0.94 [0.68,1.30] |
| Screen-based sedentary behavior (≤2 h) | 0.63 [0.56,0.71]*** | 0.69 [0.61,0.79]*** |

*P < 0.05; **P < 0.01; ***P < 0.001.
Model A: Unadjusted for covariates.
Model B: Adjusted for covariates including residential area, gender, grade, ethnicity, smoking status, chronic disease status, self-rated health, primary caregiver and highest parental education level.

adolescents who engaged in neither behavior, those who both abstained from alcohol and limited sugary drinks had a 42% lower odds of poor sleep quality (OR = 0.58, 95% CI: 0.39–0.85).

Second, a paradoxical antagonistic interaction was observed between healthy dining out and physical activity level. Compared to the reference group (low physical activity and not practicing healthy dining out), adolescents who reported both healthy dining-out habits and engaged in high levels of physical activity had a 2.81 times higher odds of poor sleep quality (OR = 2.81, 95% CI: 1.31–6.04).

## Discussion

This large-scale study among Chinese adolescents reveals that while individual healthy dietary habits are correlated with better sleep quality, the interplay between different lifestyle behaviors is complex and linked to both synergistic and counterintuitive outcomes. Our findings highlight that an integrated approach, accounting for these interactions, is essential for designing effective public health interventions.

### Independent associations: The primacy of diet

Our study demonstrated the independent association between healthy dietary patterns and better sleep quality. Behaviors such as maintaining a regular diet, having a healthy breakfast, and limiting sugary drinks were strongly associated with better sleep outcomes, which align with extensive literature (Versteeg et al., 2015; Otsuka et al., 2019). Regular meal timing helps synchronize the body's circadian rhythms (Process C), which are fundamental to regulating sleep–wake cycles (Wehrens et al., 2017). Conversely, high intake of sugar and ultra-processed foods is linked to glycemic instability and inflammation, potentially disrupting sleep architecture (Min et al., 2018). Overconsumption of calories may contribute to obesity, which is a significant risk factor for sleep disorders (Rodrigues et al., 2021; Lee and Cho, 2022). Healthy dining out can reduce the risk of poor sleep quality. Several studies (Mancino et al., 2009; Lee et al., 2016; Matsumoto et al., 2021) show that prolonged dining at restaurants is often associated with higher intake of energy and fat, but lower intake of dietary fiber, vitamin C and various micronutrients. With higher energy intake and lower diet quality, it can result in weight gain and obesity (Choi et al., 2019; Zheng et al., 2021).

The lack of a significant independent association of physical activity level, after adjusting for confounders, is noteworthy. This nonsignificant finding may suggest that its relationship with sleep is either mediated by other factors not fully captured in our model, such as psychological well-being, or that the relationship follows a nonlinear pattern where the benefits of moderate activity are offset by the potential arousal of high-intensity exertion in this specific

**Table 3.** Interaction effects analysis between dietary behavior and levels of physical activity and sedentary behavior (*N* = 15,059)

| | | Model A (95% CI) | | Model B (95% CI) | |
|---|---|---|---|---|---|
| | | OR (95% CI) | *P*-value | OR (95% CI) | *P*-value |
| **Healthy dining out** | *Physical activity level* | | | | |
| No | *Low level* | 1.00 | | 1.00 | |
| No | *Moderate level* | 0.74 [0.49,1.12] | 0.157 | 0.80 [0.52,1.23] | 0.311 |
| No | *High level* | 0.43 [0.22,0.85] | 0.014 | 0.45 [0.23,0.89] | 0.022 |
| Yes | *Low level* | 0.46 [0.41,0.52] | <0.001 | 0.58 [0.51.0.66] | <0.001 |
| Yes | *Moderate level* | 0.98 [0.59,1.62] | 0.942 | 0.933 [0.61,1.72] | 0.933 |
| Yes | *High level* | 2.70 [1.27,5.75] | 0.01 | 2.81 [1.31,6.04] | 0.008 |
| **Limited consumption of sugary beverages** | *Abstaining from alcohol* | | | | |
| No | *No* | 1.00 | | 1.00 | |
| Yes | *No* | 0.67 [0.48,0.94] | 0.021 | 0.86 [0.61,1.22] | 0.404 |
| No | *Yes* | 0.72 [0.51,1.01] | 0.055 | 0.81 [0.57,1.14] | 0.223 |
| Yes | *Yes* | 0.54 [0.37,0.78] | 0.001 | 0.58 [0.39,0.85] | 0.005 |

Model 1: Univariate analysis including only two interaction variables and their product terms.
Model 2: Adjusted for the other eight factors as confounding variables based on Model 1.

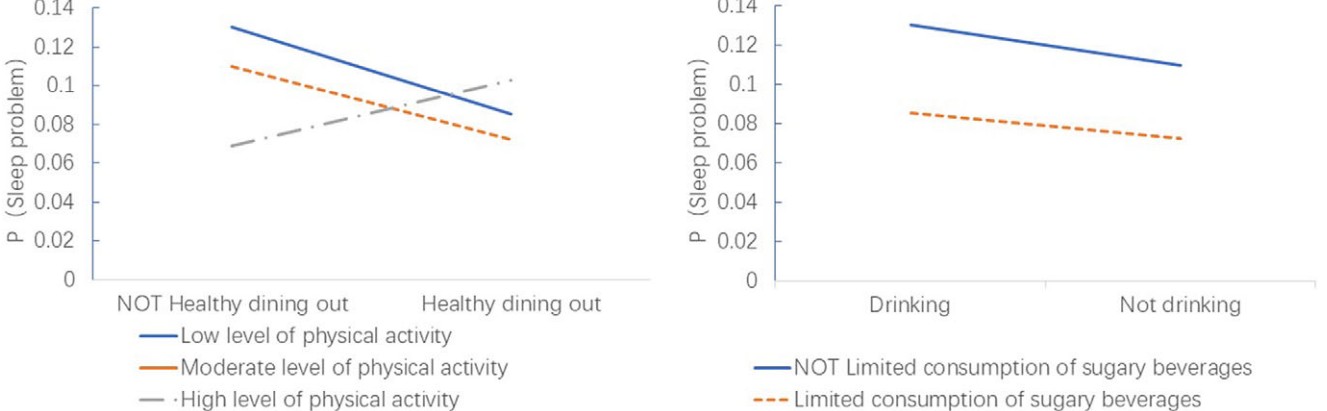

**Figure 2.** Interaction effect between *Physical activity levels* and *Healthy dining out* (A), and interaction between *Abstaining from alcohol* and *Limited consumption of sugary beverages* (B).

population. Some studies (Paluska and Schwenk, 2000; Verloigne et al., 2021) suggest that physical activity may be linked to sleep quality through its mediating role in psychology. Limiting sedentary screen time emerged as a robust protective factor, aligning with the WHO 2020 Guidelines, positioning sedentary behavior reduction as a core, independent target for health outcomes (Bull et al., 2020). However, the neurobiological impact of this behavior likely depends on its specific qualitative characteristics. As proposed in recent taxonomies of sedentary behavior, "mentally passive" screen time (*e.g.*, passive viewing) is consistently detrimental to cognitive health, whereas "mentally active" behaviors may have distinct neural correlates (Zhang et al., 2025b). Regarding temporal patterns, the lifespan brain health framework (Zou et al., 2024) and recent meta-analyses (Feter et al., 2024) underscore the cognitive necessity of interrupting sedentary bouts. Our findings thus implicate passive recreational screen exposure as a critical target; its characteristically prolonged, uninterrupted nature displaces sleep while precluding the neuroprotective benefits of frequent activity.

## Synergistic and antagonistic interactions: A deeper look into adolescent lifestyles

The most striking finding of this study is the paradoxical interaction whereby the combination of high-level physical activity and healthy dining-out habits was associated with a significantly increased risk of poor sleep quality (OR = 2.81). Unlike the additive health benefits typically assumed, this interaction suggests a "High-Load Lifestyle" pattern, where the physiological and temporal costs of maintaining these behaviors compete directly with sleep recovery.

First, interpreted through the 24-h movement behavior framework (Rollo et al., 2020), adolescent time allocation operates within a finite system where behaviors represent a zero-sum trade-off. Unlike isolated behaviors, the co-occurrence of high-load activities mandates a compositional perspective. Consistent with the "whole-day matters" paradigm, increasing time allocation to physical activity or social dining within a finite 24-h cycle necessitates isotemporal substitution, inevitably displacing sleep duration or pre-sleep quiescence. Both high-level physical activity and dining out are time-

intensive activities. Their co-occurrence imposes a substantial "time tax" on the evening schedule. In the specific context of Chinese adolescents, where academic demands already saturate evening hours (Yu et al., 2023; Fan et al., 2025), this dual burden inevitably encroaches upon the sleep window *via* isotemporal substitution (Zhang et al., 2023). Consequently, the sleep disruption is likely a function of schedule compression rather than the biological effects of the behaviors themselves (Semplonius and Willoughby, 2018).

Second, the interaction precipitates a state of heightened arousal incompatible with sleep initiation. Sleep initiation requires the down-regulation of the central nervous system. High-intensity exercise, while metabolically beneficial, induces acute physiological stress responses, including elevated core body temperature and cortisol secretion (Perrier et al., 2024). When coupled with dining out, which serves as a primary vector for adolescent social engagement, the physiological arousal of exercise is compounded by the cognitive and emotional stimulation of social interaction (Salvy et al., 2012). This synergistic arousal may delay the circadian phase and fragment sleep, overriding the nutritional advantages of "healthy" food choices.

Furthermore, the self-reported measure of "healthy dining out" may be a limitation. This term is subjective and could be interpreted differently by adolescents. In contemporary China, dining out is a major social activity for youth, ranging from quick, inexpensive meals to elaborate, trendy gatherings. An adolescent might report choosing "healthy" food options (*e.g.*, non-fried items) during a lengthy, socially stimulating dinner with friends. In this case, the negative impact on sleep would stem from the social context and timing of the meal, not the nutritional content itself. This highlights the need for future research to use more granular measures, such as time-use diaries, to disentangle the type, timing and social context of dining behaviors.

In contrast, the synergistic protective association of combining alcohol abstinence with limited sugary drink consumption is more straightforward. Both alcohol and high sugar intake are known to disrupt sleep. Alcohol fragments REM sleep, while sugary drinks can cause blood sugar spikes and subsequent crashes during the night, leading to awakenings (Smith et al., 2019; Khan et al., 2021). Eliminating both of these disruptive inputs provides a compounded benefit for sleep stability.

## Limitations and implications

The study had several limitations. First, as a cross-sectional study, it is restricted to identifying the associations rather than establishing a cause-and-effect relationship. Temporal precedence cannot be determined, and more longitudinal studies are therefore needed to provide further evidence of causality. Second, the applications of the self-administered questionnaire introduced an inevitable self-reporting bias and recall bias. Specifically, the internal consistency of the PSQI ($\alpha = 0.68$), while comparable to similar large-scale adolescent cohorts, suggests future research should integrate objective actigraphy to corroborate subjective sleep reports. Third, measurement constraints limit the interpretation of specific behavioral variables. While our "Screen-based sedentary behavior ($\leq 2$ h)" metric effectively captures recreational sedentary behavior, it excludes the extensive educational sitting inherent to the Chinese academic context. Similarly, the operationalization of "healthy dining out" captured dietary quality but failed to quantify social context or temporal duration. Finally, unmeasured confounding remains a substantial constraint. Future research should incorporate these variables to build a more comprehensive explanatory model.

Despite these limitations, this study has several theoretical and practical contributions. From a theoretical perspective, this study contributes to the existing literature by elucidating the complex, nonlinear interactive associations between dietary behaviors and physical activity on sleep quality. From a practical perspective, given the large sample size covering a representative county, this study had a large sample size, which included all middle schools in the northeast county of China. Practically, these findings advocate for holistic interventions that move beyond isolated behavioral promotion. Effective strategies must address the temporal trade-offs adolescents face, guiding them to balance the scheduling of academics, athletics and social dining to mitigate physiological hyperarousal and protect sleep windows.

## Conclusion

This study suggests that healthy dietary behaviors are positively associated with sleep quality among adolescents. More significantly, it uncovers the complex, nonlinear interactions between lifestyle factors. The finding that combining two ostensibly healthy behaviors – high physical activity and healthy dining out – can paradoxically be associated with an increase in the risk of poor sleep provides a critical insight for public health. It suggests that interventions should not simply promote individual healthy behaviors in isolation. Instead, a holistic, 24-h approach is needed, guiding adolescents on how to balance and schedule their activities – academics, exercise, socializing and rest – to avoid antagonistic outcomes and protect their sleep. Educators, parents and policymakers must recognize that promoting adolescent health requires an understanding of these intricate behavioral trade-offs.

**Open peer review.** To view the open peer review materials for this article, please visit http://doi.org/10.1017/gmh.2026.10138.

**Data availability statement.** The national dataset used in this research contains personal identifiable information, which was used to link test results and is not available to the public. Any data requests should be made to the corresponding author.

**Acknowledgements.** The authors wish to thank all those who kindly volunteered to participate in the study.

**Author contribution.** B.H.: Conceptualization, methodology, formal analysis, data curation, writing – original draft and visualization.
N.H.: Conceptualization, methodology and investigation.
C.C.: Conceptualization, methodology and investigation.
J.G.: Funding acquisition, project administration, supervision, writing – review and editing. All authors contributed to the refinement, proofreading and double-checking of the figures in the final draft. All authors have read and approved the final manuscript for submission.

**Financial support.** This work was supported by the Beijing Municipal Social Science Foundation (grant number: 22JYA002). The funders had no role in study design, data collection and analysis, decision to publish or preparation of the manuscript.

**Competing interests.** The authors declare none.

**Ethical consideration.** Before data collection, this project received ethical approval from the Peking University Biomedical Institutional Review Board (No. IRB00001052–24064). All participants and their guardians provided written informed consent after being briefed on the study's purpose.

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
