## [Reviewer Report]

Thank you for the opportunity to review this important and well-executed study. The manuscript presents fascinating findings on a topic of great public health relevance. I commend the authors on their large sample size and sophisticated analysis of interaction effects, which yields truly interesting results.

To further strengthen the manuscript, I would be grateful if the authors could clarify a few points for me:

1.Regarding the measurement of sleep quality, the Cronbach’s alpha for the PSQI in this sample is reported as 0.68. Could the authors please comment on this value, as it is somewhat lower than the thresholds commonly cited for established scales?

2.The operational definition of physical activity intensity is based on the duration and frequency of exercise. Given that nearly all Chinese middle school students participate in mandatory Physical Education (PE) classes, could the authors help me understand how this method distinguishes between students who only participate in PE and those who engage in additional voluntary exercise? I was wondering if this might influence the interpretation of the ‘low activity’ category.

3.The introduction effectively sets the stage for the study. However, I was curious if the authors considered exploring the existing literature on the potential for a non-linear (e.g., inverted U-shape) relationship between physical activity intensity and sleep outcomes? A slightly more detailed synthesis of the established mechanisms and complexities in the PA-sleep relationship might provide an even stronger theoretical foundation for the intriguing interaction effects found later.

4.Similarly, regarding the literature review, I was wondering if a more detailed summary of the key established findings and consensus views on both diet-sleep and PA-sleep relationships could be integrated to further contextualize the study’s hypotheses?

Thank you again for sharing your valuable work. I look forward to seeing the authors' responses.

---

## [Reviewer Report]

Overall, Strength and Significance

The study’s primary strength lies in its very large sample size (N=15,059), which provides high statistical power and enhances the generalizability of findings within the target population (Chinese middle school students). The use of validated instruments like the Pittsburgh Sleep Quality Index (PSQI) ensures the reliability of the outcome measure. Most significantly, the research moved beyond simple independent associations to explore behavioral interactions, which is critical for developing real-world, integrated public health interventions.

General comment

Methodology

The most critical limitation, inherent to the design, is the cross-sectional nature of the data. While the study suggests associations, it cannot establish causality. For example, poor sleep quality might lead to unhealthy dietary choices, rather than the diet causing the poor sleep. This limitation should be explicitly acknowledged when discussing the protective nature of the identified behaviors.

Result

The finding that the combination of healthy dining-out practices and high physical activity was associated with 181% increased odds of poor sleep quality (OR=2.81) is highly counterintuitive and requires a more rigorous discussion. While the conclusion correctly labels this antagonistic, the mechanism suggested by the term “antagonistic” (counter-productive) is insufficient. Potential concerns include:

1. High physical activity, especially if coupled with frequent dining out, might simply reflect a highly time-constrained lifestyle, leading to later bedtimes or sleep displacement rather than a true biological antagonism between the two behaviors.

2. Adolescents engaged in intense, high-level sports and who need to carefully manage diet/social life might be under higher academic or athletic stress, which is the true driver of poor sleep.

3. Definition of “Healthy Dining-Out”: The scale used for this variable need’s scrutiny. If it measures frequency rather than quality of choices, increased dining-out frequency might simply indicate more late-night social activities that cut into sleep.

Discussion

The Discussion section requires careful revision to maintain consistency with the cross-sectional study design. Specifically, language used in sections, such as Section 4.1 on page 10, must be modified to avoid implying causality. Terms suggesting cause-and-effect should be replaced with language that denotes only association, correlation, or relationship.

---

## [Editor Report]

Dear Authors,

Thank you for submitting your manuscript to Cambridge Prisms: Global Mental Health. We appreciate your efforts in addressing an important topic related to lifestyle behaviors and sleep health among adolescents. Your study benefits from a very large sample size and the analysis of behavioral interactions offers valuable potential insights for global mental health promotion. However, after careful evaluation of both the manuscript and the detailed peer review feedback, we believe that major revisions are necessary before the manuscript can be considered for publication.

Below we summarize key issues raised by the reviewers that require your attention:

1.Clarifications in Measurement and Operationalization 

Reviewer 1 requests clarification on important methodological points: 

• The relatively low Cronbach’s alpha for the PSQI (α = 0.68) should be justified with references to prior research and potential sample-specific characteristics. 

• The definition of physical activity intensity may not adequately differentiate mandatory school PE from voluntary exercise, influencing interpretation of the “low activity” category.

Please expand your methodological justification and acknowledge potential limitations.

2.Strengthening the Literature Foundation 

Reviewer 1 recommends enhancement of the introduction and rationale: 

• Include discussion of potential non-linear relationships (e.g., inverted U-shaped) between physical activity and sleep outcomes. 

• Provide a more detailed synthesis of well-established evidence on both diet–sleep and PA–sleep relationships.

This will help ground your hypotheses more robustly in prior theory.

3.Explain the Counterintuitive Interaction Effect 

Reviewer 2 highlights that the finding of an antagonistic effect — high PA combined with healthy dining-out behavior associated with increased odds of poor sleep (OR = 2.81) — requires deeper scientific interpretation: 

• Consider potential confounding factors such as time pressure, late-night social or sport commitments, or stress from competitive activities. 

• Revisit the definition and scoring of “healthy dining-out” given its possible relationship with social/nighttime behavior rather than nutritional quality.

A more comprehensive interpretation will strengthen the credibility of this novel finding.

4. Modify Language to Avoid Implied Causality 

Reviewer 2 emphasizes that the cross-sectional design does not support causal inference: 

• Please revise all statements suggesting cause-and-effect to use terms such as association, relationship, or correlation. 

• Ensure consistency across the Abstract, Discussion, and Conclusion sections.

This adjustment is essential to maintain methodological accuracy.

5.Additional Editorial Recommendations 

To further enhance clarity and rigor: 

• Expand the Limitations section (measurement constraints, unmeasured stress variables, self-report bias). 

• Provide justification for categorizations of physical activity and dietary behaviors. 

• Improve focus and flow in the Discussion to highlight the public health implications for adolescent sleep and mental well-being.

Editorial Decision: Major Revision Required 

We invite you to revise your manuscript carefully in response to the above points and the detailed reviewer feedback. When resubmitting to Cambridge Prisms: Global Mental Health, please include:(1) A comprehensive point-by-point response document; (2) A tracked-changes version of the manuscript highlighting all text modifications

We believe that, with these substantial revisions, your work has strong potential to contribute meaningfully to global mental health research and intervention strategies for youth.

Thank you again for choosing Cambridge Prisms: Global Mental Health. We look forward to receiving your revised submission.

With best regards, 

Dr. Liye Zou 

Associate Editor 

Cambridge Prisms: Global Mental Health

---

## [Reviewer Report]

Regarding the submitted manuscript, the following comments are put forward concerning the reference citation section:

Issue of Unranslated Cited Literatures

There are some cited literatures in the manuscript that have not been translated. It is recommended to supplement and complete the corresponding translation work:

For English cited literatures in a Chinese manuscript, it is necessary to add Chinese translations of the literature titles and Chinese paraphrases of the core contents of the literatures, so as to facilitate Chinese readers to understand the core value of the literatures.

For foreign language literatures cited that are not in Chinese or English, corresponding Chinese translation contents should also be supplemented.

Reminder for Confirming the Format of Literature Contents

Please refer to the literature citation format specifications required by the manuscript (such as GB/T 7714, APA, MLA and other corresponding format standards) and confirm the following contents one by one:

Whether the bibliographic items of the cited literatures are complete, including author(s), literature title, journal name/book name, publication year, volume/issue, page numbers, etc.

Whether the corresponding relationship between citation marks and the list of references at the end of the manuscript is accurate.

Whether the formats of different types of literatures (journal papers, dissertations, monographs, conference papers, etc.) meet the requirements.

---

## [Editor Report]

Dear authors, 

Thanks for your careful revision. Before accepting this manuscript, Additional editorial comments for the revision. Your manuscript already treats lifestyle factors as interacting and time-competing behaviors (including screen-based sedentary behavior and a “24-hour approach”). However, the Discussion needs a substantially deeper treatment of sedentary behavior and the broader “24-hour movement behavior” framework (physical activity, sedentary behavior, and sleep as co-dependent parts of a finite day). Please revise the Discussion to include the following:

1.Position sedentary behavior as a core guideline target (not a secondary covariate).

The 2020 WHO Guidelines explicitly recommend reducing sedentary behaviors across age groups, and for children/adolescents emphasize limiting sedentary time (with particular concern for recreational screen time), even though evidence was insufficient to specify a single universal sedentary threshold. In this context, please interpret your “≤2h electronic-device sedentary behavior” measure more carefully (what it captures and what it misses, including total sitting and domain/context). Bull, F. C., Al-Ansari, S. S., Biddle, S., Borodulin, K., Buman, M. P., Cardon, G., Carty, C., Chaput, J. P., Chastin, S., Chou, R., Dempsey, P. C., DiPietro, L., Ekelund, U., Firth, J., Friedenreich, C. M., Garcia, L., Gichu, M., Jago, R., Katzmarzyk, P. T., Lambert, E., … Willumsen, J. F. (2020). World Health Organization 2020 guidelines on physical activity and sedentary behaviour. British journal of sports medicine, 54(24), 1451–1462. https://doi.org/10.1136/bjsports-2020-102955

2. Discuss the time-use constraint and “trade-offs” explicitly (24-hour movement behaviors).

Because increasing time in one behavior necessarily displaces time in another, associations with sleep and brain health should be framed as time reallocations (isotemporal substitution/compositional thinking), not independent effects. The “whole-day matters” paradigm and low adherence to integrated 24-hour recommendations should be acknowledged. This framing is also important when interpreting any counterintuitive patterns (for example, physically active adolescents may also have schedules, training times, or stressors that affect sleep). Rollo, S., Antsygina, O., & Tremblay, M. S. (2020). The whole day matters: Understanding 24-hour movement guideline adherence and relationships with health indicators across the lifespan. Journal of sport and health science, 9(6), 493–510. https://doi.org/10.1016/j.jshs.2020.07.004

3. Add nuance: not all sedentary behavior is equivalent for brain health.

Emerging work emphasizes that sedentary behavior effects can be context-dependent (mentally passive vs mentally active; content and cognitive engagement), which is directly relevant to interpreting “screen time” measures. Zhang, Z., Chen, Y., Yu, Q., Li, J., Zou, L., Mavilidi, M. F., Green, C. S., Owen, N., Hallgren, M., Raichlen, D., Lu, S., Alexander, G. E., Paas, F., & Herold, F. (2025). A neurobiological taxonomy of sedentary behavior for brain health. Trends in neurosciences, 48(11), 853–864. https://doi.org/10.1016/j.tins.2025.09.002

Please integrate this nuance to avoid implying that all sedentary time is uniformly harmful or that reducing screen time alone fully captures the sedentary-behavior construct.

4. Integrate evidence on sedentary behavior, activity breaks, and brain outcomes across the lifespan (as context, not as a new aim). Briefly summarize that (i) sedentary behavior has been linked to brain health outcomes across ages (Zou, L., Herold, F., Cheval, B., Wheeler, M. J., Pindus, D. M., Erickson, K. I., Raichlen, D. A., Alexander, G. E., Müller, N. G., Dunstan, D. W., Kramer, A. F., Hillman, C. H., Hallgren, M., Ekelund, U., Maltagliati, S., & Owen, N. (2024). Sedentary behavior and lifespan brain health. Trends in cognitive sciences, 28(4), 369–382. https://doi.org/10.1016/j.tics.2024.02.003), and (ii) interrupting prolonged sitting with physical activity shows measurable cognitive/brain-relevant effects in experimental and synthesis work (Effects of reducing sedentary behaviour by increasing physical activity, on cognitive function, brain function and structure across the lifespan: a systematic review and meta-analysis). 

5. Terminology and conceptual clarity.

Please use consistent terminology for the integrated construct (physical activity, sedentary behavior, sleep) and acknowledge ongoing efforts to standardize language in this rapidly growing area. Falck, R. S., Davis, J. C., Li, L., Stamatakis, E., & Liu-Ambrose, T. (2022). Preventing the ‘24-hour Babel’: the need for a consensus on a consistent terminology scheme for physical activity, sedentary behaviour and sleep. British journal of sports medicine, 56(7), 367–368. https://doi.org/10.1136/bjsports-2021-104487